# New Imaging Features of Multinodular and Vacuolating Neuronal Tumor Revealed by Alcohol and Illicit Drugs Consumption

**DOI:** 10.3390/diagnostics12112779

**Published:** 2022-11-14

**Authors:** Carmen Adella Sirbu, Constantin Ștefani, Sorin Tuță, Aida Mihaela Manole, Octavian Mihai Sirbu, Raluca Ivan, Gabriela Simona Toma, Alexandra Giorgiana Calu, Dragos Catalin Jianu

**Affiliations:** 1Department of Neurology, ‘Dr. Carol Davila’ Central Military Emergency University Hospital, 010242 Bucharest, Romania; 2Department of Family Medicine and Clinical Base, ‘Dr. Carol Davila’ Central Military Emergency University Hospital, 051075 Bucharest, Romania; 3Department No. 5, ‘Carol Davila’ University of Medicine and Pharmacy, 050474 Bucharest, Romania; 4Clinical Neurosciences Department, ‘Carol Davila’ University of Medicine and Pharmacy “Carol Davila”, 050474 Bucharest, Romania; 5Department of Neurology, National Institute of Neurology and Neurovascular Diseases, 041914 Bucharest, Romania; 6Neurosurgery Department, ‘Dr. Carol Davila’ Central Military Emergency University Hospital, 010242 Bucharest, Romania; 7Department of Radiology, ‘Dr. Carol Davila’ Central Military Emergency University Hospital, 010242 Bucharest, Romania; 8Department of Pathology, ‘Dr. Carol Davila’ Central Military Emergency University Hospital, 010242 Bucharest, Romania; 9Department of Neurology, Victor Babes University of Medicine and Pharmacy, 300041 Timisoara, Romania

**Keywords:** Multinodular and vacuolating neuronal tumor (MVNT), dysembryoplastic neuroepithelial tumors (DNET), leave me alone, MRI, seizure, generalized tonic-clonic seizure, illegal substances, spectroscopy, diffusion tensor imaging (DTI), tractography

## Abstract

It has been almost a decade since the multinodular and vacuolating neuronal tumor (MVNT) was first described. In 2021, WHO classified it as a defined entity, and it is considered one of the glioneuronal and neuronal tumors. Due to its similarities with dysembryoplastic neuroepithelial tumors (DNET), some authors consider it a variant of these, ranking in the category of malformations, but genetic alterations favor a neoplastic origin. We present a 29-year-old male with a generalized onset tonic-clonic seizure after a nightclub party. Imaging studies revealed a right temporal multinodular and vacuolating neuronal tumor confirmed by biopsy. It is considered a nonaggressive, “leave me alone” brain lesion, which does not require biopsy because of well-defined MRI characteristics. Surgery is indicated only in symptomatic cases. We consider that this lesion was revealed by his seizure, most probably provoked (with normal video EEG recording) by the consumption of a lot of alcohol, illicit drugs, and sleep loss after a club party. We recommended close monitoring, but our patient preferred the surgery. Our case added more imaging details corroborated with the histopathology features of MVNT. FLAIR images revealed hypointense nodules surrounded by hyperintense peripheral rings and areas of high signal intensity between the nodules, which correspond to the histopathological architecture. To our knowledge, this is the first case of MVNT with diffusion tensor imaging and fiber tractography imaging studies.

A 29-year-old male smoker and drug addict (over the last 10 years) presented a generalized onset seizure from the family description. It appeared in the context of 25 h of sleep deprivation along with the consumption of alcohol, energy drinks, cocaine, and cigarettes after a night of partying. He had a seizure-free history and no afflictions or chronic medication, except for a depressive episode one year prior, an episode for which he was treated with antidepressants. His medical background was not relevant. Any neurological deficits were absent. No seizure history was found in his family, and no other medical conditions were found until this event. Lab tests were normal. The 30 min EEG monitoring was normal.

The brain CT and conventional and advanced MRI features were appropriate for a multinodular and vacuolating neuronal tumor (MVNT).

The CT scan shows a hypodense cortical/subcortical white-matter lesion mainly on the right temporal lobe with partially lobulated margins, such as a cluster of hypodense nodules (Figure 1). 

On the MRI exam, the patient has an extensive lesion in the right temporal lobe at the anterior pole, inferior, and medial temporal gyrus, extended to the amygdala and hippocampus. The lesion exhibits a non mass appearance with imaging findings consisting of a cluster of nodular lesions, gathered on the inner surface of the cortex at the cortical-subcortical interface. Some of the lesions are arranged tightly together, but some of them are close by and have a small subcortical white matter interface. Furthermore, we have analyzed the imaging findings and tried to correlate them with the data from the literature regarding the imaging and histopathological characteristics of the tumor [1,2,3]. On the T1wi, the tumor has small hypointense, homogeneous, and well defined nodules, located at the cortical-subcortical junction, more developed towards the cortical side. Some of them are clustered tightly together, and some preserve a small subcortical and cortical interface between them (Figure 2).

The T2wi displays an extensive lesion that has a bunch of grape appearances with lobulated margins and high signal intensity. In this sequence, the nodules seem to be more closely together than on the T1wi, with no normal parenchyma interface between their components (Figure 3 and Figure 4). The scattered nodules are well defined and appear to be homogeneous.

The tumor does not exhibit restriction of diffusion on DWI and ADC (Figure 4).

One of the key sequences is T2 space inversion recovery with fluid attenuation (T2 FLAIR), which has fine slices (in our case 1.3 mm slice) and identifies the clustered and the scattered nodules, but with an interesting characteristic. The larger nodules have a hypointense center with a hyperintense rim with a ring-like appearance, and between the nodules, there are areas with a high signal intensity that connect the nodules. The smaller nodules have nonhomogeneous high signal intensity with a central dot of low signal intensity (Figure 5).

Therefore, the question is why it has this appearance (Figure 5). Searching the histological analysis, we found that this kind of tumor has stromal vacuolation of the neuroepithelial cells arranged in nodules in the deep cortical ribbon and superficial subcortical white matter. The vacuolation was noted in the cytoplasm and also in the pericellular region. In some papers, a higher content of synaptophysin, which is a glycoprotein, was noted. Also, surrounding the affected neurons were found unmyelinated nerve fibers.

Combining the signals on T1, T2, and T2 FLAIR, and with the knowledge gathered, we concluded that the signal intensity of the nodules is nonhomogeneous because of the vacuoles that are low T1 high T2 and low on FLAIR. The periphery of the nodules has high signal intensity on FLAIR because of the high protein content and the demyelination of the neurons, and the areas surrounding the nodules have high signal intensity on FLAIR given by the unmyelinated nerve fibers. 

MRI spectroscopy shows an increase in the choline peak and a decrease in the N-acetyl aspartate peak (Figure 6A). Tractography revealed partial disruption of the normal structure due to the absence of temporal lobe cortical layers in the affected area. (Figure 6B,C). 

Because we did not have enough data on the semiology of the seizure, we advised him to take anticonvulsant treatment according to the ILAE recommendation from 2017. Studies show that these lesions occur during neurogenesis [1]. The most correct attitude is to carefully monitor. Our patient preferred the surgery, and the histopathologic features revealed a multinodular and vacuolating neuronal tumor.

Usually, MVNT is diagnosed due to adult-onset refractory epilepsy, headache related incidents, or incidentally. Especially in childhood, MVNT could be associated with focal cortical dysplasia [1]. A series of 33 cases conclude that MVNTs are benign, non-aggressive lesions that do not require biopsy in asymptomatic patients and behave more likely as a malformation process than a true neoplasm [1]. Debate on this lesion’s neoplastic or malformative character is still present, but genetic alterations favor a neoplastic origin [4,5]. The last WHO tumor brain classification from 2021 rated MVNTs as grade I, with the mitogen-activated protein kinase (MAPK) as a molecular marker [6]. 

Brain CT is usually normal or shows a non-enhancing/non-calcifying cortical/subcortical white-matter hypodense lesion or asymmetry [7,8]. MRI lesions are described as multiple clustered nodules in the cortical–subcortical white matter, without peripheral edema or mass effect. The lesion is usually hypointense on T1, hyperintense on T2-weighted sequences, and hyperintense on FLAIR (nonhomogeneous in our case with hypointense nodules included), without restriction on diffusion-weighted (DWI), and no susceptibility on susceptibility-weighted imaging (SWI) or gradient recalled echo (GRE) sequences [9,10]. No contrast enhancement was found. In our case, MRI spectroscopy showed a slight increase in choline (Cho) and a reduction of N-acetyl aspartate (NAA) [11]. In many cases, positron emission tomography revealed an increase in 11C-methionine uptake in the lesion [12,13]. Therefore, sometimes, this lesion is confused with low-grade glioma [1]. During surgery, it was not possible to determine the obvious limit between the tumor and the brain [14]. Histopathologic studies show dysplastic cells perpendicular to the cortical surface, that exhibit immunopositivity for HuC/HuD and Olig2. This is proof that MVNT appears at an earlier stage of neuronal development, during neurogenesis [1].

One of the difficulties we encountered was the differentiation between a single situation-related seizure and the onset of structural epilepsy. What may support that the diagnosis in the first scenario was the generalized onset, from the family description, and the context in which it occurred—sleep deprivation and substance use. In this situation, we can say that the tumor was discovered incidentally. On the other hand, in structural epilepsy, seizures are focal with or without lateralization, therefore possibly with temporal lobe semiology. This is difficult to prove in this hypothesis through a semiology based on the relatives’ statements, a short EEG monitoring, and with a variety of possible triggers.

Therapeutic management can be achieved with antiepileptic drugs or surgery. Drug treatment has the advantage of being noninvasive, but with the risk of potential side effects. The possible adverse effects, and the fact that its administration is required throughout the entire life, were the main disadvantages for our patient. At the same time, surgical therapy may have the chance of complete remission of epileptic seizures and confirmation of the diagnosis based on histological analysis, which was only a probable diagnosis on imaging criteria, but with the possible disadvantage of presenting postoperative seizures or deficits on the surgical scar. 

To conclude, we consider that this tumor was discovered due to a generalized onset seizure triggered by multiple factors, and the most correct attitude would have been to carefully monitor. This is a rare case found in medical practice and is also rarely described in the literature. Our case added more imaging details corroborated with the histopathology features of MVNT. FLAIR images revealed hypointense nodules surrounded by hyperintense peripheral rings and areas of high signal intensity between the nodules, which correspond to the histopathological architecture. Additionally, to our knowledge, it is the first case in which spectroscopy and fiber tractography aspects were analyzed, excluding a high-grade tumor, and this confirmed the disruption of the temporal stem fibers.

## Figures and Tables

**Figure 1 diagnostics-12-02779-f001:**
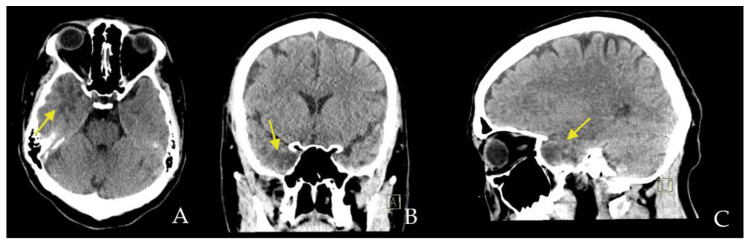
Brain CT in axial (**A**), coronal (**B**), and sagittal (**C**) planes reveals a hypodense area in the right temporal lobe with cortical and subcortical topography (yellow arrow), with partially lobulated contour, such as a cluster of hypodense nodules.

**Figure 2 diagnostics-12-02779-f002:**
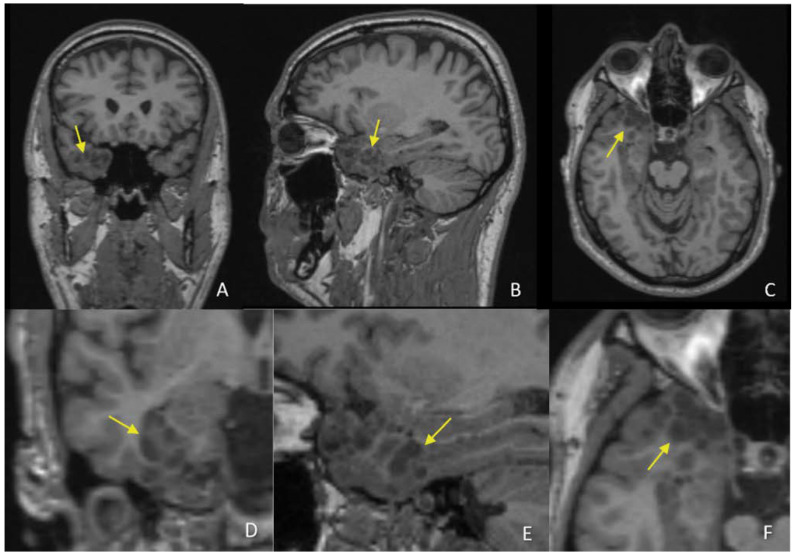
T1wi in coronal (**A**), sagittal (**B**), and axial (**C**) planes with correspondent zoomed images (**D**–**F**) on the temporal lesion reveal hypointense nodules (yellow arrow) at the cortical–subcortical white matter junction, some of them being clustered together.

**Figure 3 diagnostics-12-02779-f003:**
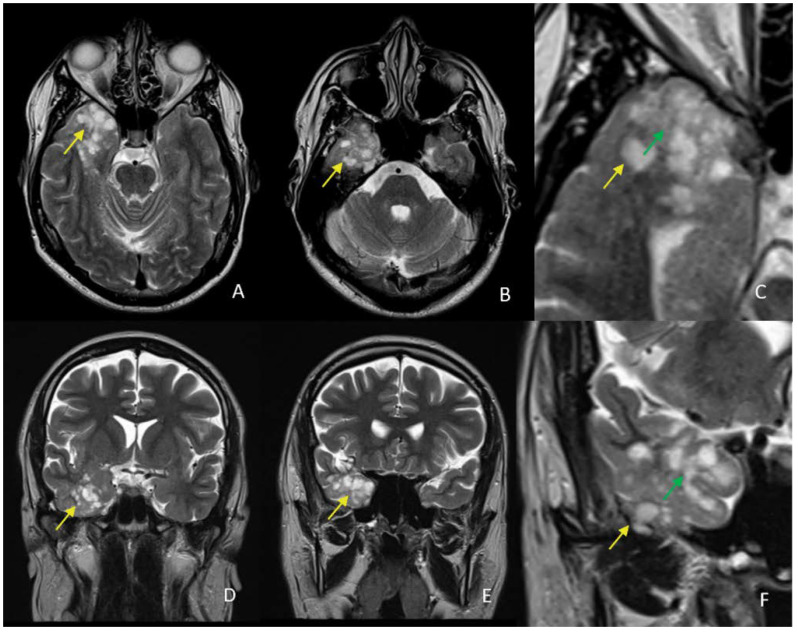
T2wi in axial (**A**,**B**) and coronal (**D**,**E**) planes with correspondent zoomed images (**C**,**F**) on the temporal lesion reveals hyperintense nodules (yellow arrow) at the cortical–subcortical white matter junction with areas of high signal intensity between them (green arrow).

**Figure 4 diagnostics-12-02779-f004:**
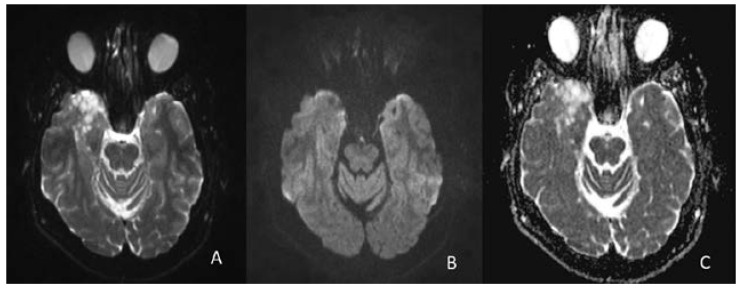
DWI (**A**,**B**) and ADC (**B**) in axial planes reveal the lesion with high signal intensity on DWI at b0 (**A**) and isointense at b800 (**B**) with high signal intensity on ADC (**C**), without restricted diffusion.

**Figure 5 diagnostics-12-02779-f005:**
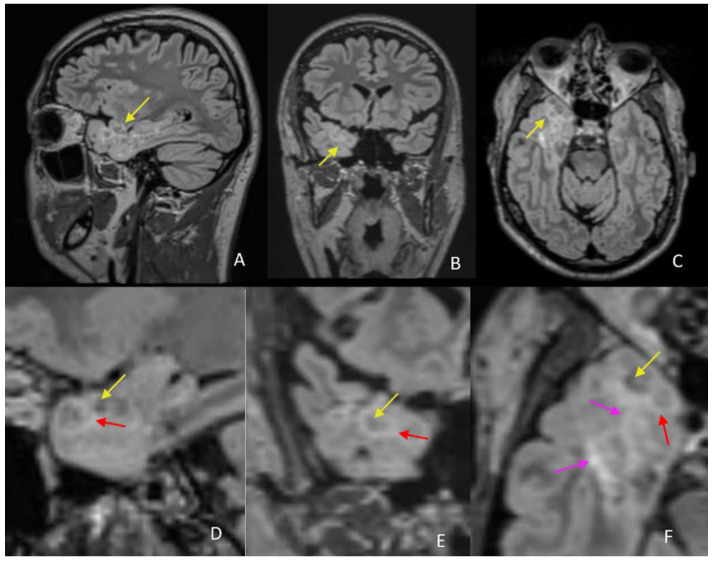
T2 FLAIR in sagittal (**A**), coronal (**B**), and axial (**C**) planes with correspondent zoomed images (**D**–**F**) on the temporal lesion reveals hypointense nodules at the cortical–subcortical white matter junction (yellow arrow) with a peripheral high signal intensity ring (red arrow) and areas of high signal intensity between them (pink arrow).

**Figure 6 diagnostics-12-02779-f006:**
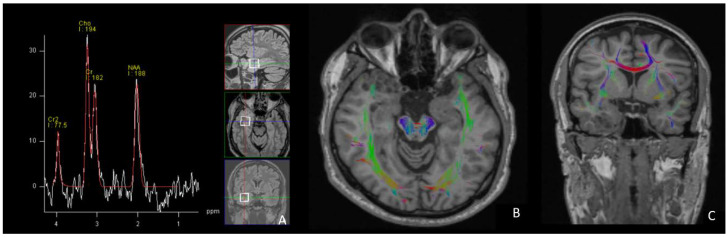
MR spectroscopy (**A**) shows a high choline peak and a lower N-acetyl aspartate peak. Tractography in axial (**B**) and coronal (**C**) planes shows disruption of the temporal stem fibers.

## Data Availability

All relevant data have been presented in this manuscript, and further inquiry can be directed to the corresponding author.

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
