# Peer review of "New Imaging Features of Multinodular and Vacuolating Neuronal Tumor Revealed by Alcohol and Illicit Drugs Consumption"

_diagnostics, 2022, doi:10.3390/diagnostics12112779_

Round 1

Reviewer 1 Report (Previous Reviewer 1)

I feel the manuscript incorporated my previous comments and has been significantly improved.

Reviewer 2 Report (Previous Reviewer 2)

The authors have answered to my queries and have modified the manuscript accordingly. I am satisfied with the revised version.

This manuscript is a resubmission of an earlier submission. The following is a list of the peer review reports and author responses from that submission.

Round 1

Reviewer 1 Report

In this manuscript, the authors reported a young adult case with MVNT, in which a generalized tonic-clonic seizure after substance uses and sleep disturbance at the night party was the chance to find the tumor. The MRI showed some typical findings of MVNT, which was pathologically confirmed, and the initial seizure was considered more likely to be a provoked one.

While MVNT is a newer type of tumor, there are already clinical reports of MVNT, which seems consistent with this case. The novelty of the current case may be the issue of night party and the DTI analysis. However, the clinical meaning of the former is quite unclear, and the latter was only presented without discussing its significance. Thus, I didn’t feel the current case is worth publishing. Other specific comments are below.

-       Title. Why only included “cocaine consumption” despite the use of other substances, e.g., alcohol.

-       Line 44. Did the generalized seizure accompany some focal features, e.g., left/right difference? If it was primary GTCS, it should support the provoked one. If it was focal-to-bilateral tonic clonic seizure, it might originate from focal lesion, possibly the tumor.

-       Line 51, the EEG monitoring. How long was the EEG monitoring performed? Was that routine EEG, e.g., for 30 min? Or long-term monitoring for several days? Only single routine EEG is not enough to rule out epilepsy.

-       Line 64, “Our patient preferred the surgery, and the histopathologic features revealed a multinodular and vacuolating neuronal tumor.” What does “the surgery” mean? Is it biopsy or resection surgery?

Reviewer 2 Report

The authors describe the case of a patient with an isolated seizure and a multi nodular and vacuolating neuronal tumor. English language needs thorough revision.

Moreover, a number of points should be modified:

- line 44-45: according to current ILAE classification, the seizure should be described as generalized onset tonic-clonic seizure.

- line 53: what is a "cluster" appearance at CT scan?

- line 61: why did the authors choose to prescribe anti seizure medication (and not "anticonvulsant drug" as stated) to a patient with a single situation-related seizure?

- line 81: was molecular study for MAPK performed in this patient?

- (more general comment regarding the whole case report): the differential diagnosis between single situation-related seizure and onset of a structural epilepsy must be elucidated. The authors should discuss the points in favor of these two diagnoses and the advantages/disadvantages of the different treatments (none, anti seizure medication or surgery).